# Therapeutic Potential of P110 Peptide: New Insights into Treatment of Alzheimer’s Disease

**DOI:** 10.3390/life13112156

**Published:** 2023-11-02

**Authors:** Ankita Srivastava, Maryann Johnson, Heather A. Renna, Katie M. Sheehan, Saba Ahmed, Thomas Palaia, Aaron Pinkhasov, Irving H. Gomolin, Joshua De Leon, Allison B. Reiss

**Affiliations:** Department of Medicine and Biomedical Research Institute, NYU Grossman Long Island School of Medicine, Mineola, NY 11501, USA; ankita.srivastava@nyulangone.org (A.S.); maryann.johnson@nyulangone.org (M.J.); heather.renna@nyulangone.org (H.A.R.); katie.sheehan2@nyulangone.org (K.M.S.); saba.ahmed@nyulangone.org (S.A.); thomas.palaia@nyulangone.org (T.P.); aron.pinkhasov@nyulangone.org (A.P.); irving.gomolin@nyulangone.org (I.H.G.); joshua.deleon@nyulangone.org (J.D.L.)

**Keywords:** mitochondria, Drp1, amyloid beta, mitochondrial fragmentation, Alzheimer’s disease, neuron, oxidative stress

## Abstract

Mitochondrial degeneration in various neurodegenerative diseases, specifically in Alzheimer’s disease, involves excessive mitochondrial fission and reduced fusion, leading to cell damage. P110 is a seven-amino acid peptide that restores mitochondrial dynamics by acting as an inhibitor of mitochondrial fission. However, the role of P110 as a neuroprotective agent in AD remains unclear. Therefore, we performed cell culture studies to evaluate the neuroprotective effect of P110 on amyloid-β accumulation and mitochondrial functioning. Human SH-SY5Y neuronal cells were incubated with 1 µM and 10 µM of P110, and Real-Time PCR and Western blot analysis were done to quantify the expression of genes pertaining to AD and neuronal health. Exposure of SH-SY5Y cells to P110 significantly increased APP mRNA levels at 1 µM, while BACE1 mRNA levels were increased at both 1 µM and 10 µM. However, protein levels of both APP and BACE1 were significantly reduced at 10 µM of P110. Further, P110 treatment significantly increased ADAM10 and Klotho protein levels at 10 µM. In addition, P110 exposure significantly increased active mitochondria and reduced ROS in live SH-SY5Y cells at both 1 µM and 10 µM concentrations. Taken together, our results indicate that P110 might be useful in attenuating amyloid-β generation and improving neuronal health by maintaining mitochondrial function in neurons.

## 1. Introduction

Alzheimer’s disease (AD) is one of the most prevalent neurodegenerative diseases and the most common cause of dementia. It is characterized by progressive memory loss and cognitive decline [1,2]. AD is often diagnosed in older persons, and an estimated 6.7 million Americans older than age 65 have been diagnosed with AD [3]. AD is associated with the accumulation of amyloid-β (Aβ) plaques outside neurons and neurofibrillary tangles (NFTs) of the protein tau within neurons [4,5,6]. Aβ plaques consist of deposition of misfolded Aβ peptide, a cleavage product of amyloid precursor protein (APP) [7]. NFTs are the aggregation of hyperphosphorylated tau, a microtubule-associated protein. It has long been assumed that the accumulation of Aβ plaques and NFTs causes neuronal dysfunction and death, leading to dementia in patients with AD [6,8]. However, newer studies have led to doubts about the causal relationship between Aβ and AD, with more and more researchers acknowledging the possibility that Aβ plaques may be biomarkers that indicate a disease process while not the direct cause of cytotoxicity [9,10,11].

Cellular metabolic changes, such as disruption of insulin signaling, occur within the AD brain and are involved in Aβ and NFT-induced toxicity [12,13]. Accumulation of both Aβ and tau protein has detrimental effects on mitochondrial functioning [14,15]. Mitochondrial dysfunction involves reactive oxygen species (ROS) generation, disruption in the electron transport chain, calcium dyshomeostasis, and apoptosis [16,17]. Mitochondrial dynamics play an essential role in regulating metabolism across the brain, the impairment of which leads to the pathogenesis of many neurodegenerative diseases, including AD [18,19]. Excessive mitochondrial fission and reduced fusion lead to mitochondrial fragmentation, which has been observed in animal models and patients with AD [20,21,22]. Structural damage to mitochondria is responsible for reduced ATP production and more ROS generation with neuroinflammation and toxicity for brain cells [23]. Although many studies focused on the effect of mitochondrial dysfunction in the development of AD, the exact mechanism is still poorly understood. 

An adequate in vitro model is fundamental for preclinical studies of the pathophysiological mechanism of disease and for the assessment of the impact of potential therapies. The most common in vitro model used to study neuronal development is the human neuroblastoma SH-SY5Y cell line [24,25]. The SH-SY5Y cell line is a subline of the neuroblastoma SK-N-SH cell line. These cells differentiate into neuronal-like cells and display biochemical and morphological features similar to mature neurons, thus highly resembling mature human neurons [24].

P110 is a seven amino acid peptide that acts as a dynamin-1-related protein (Drp1) inhibitor and thus hampers mitochondrial fragmentation [26]. Drp1 is a member of the dynamin family of GTPases, which serve as master regulators of mitochondrial fission [27]. Under normal conditions, Drp1 is located in the cytosol but translocates from the cytosol to the outer mitochondrial membrane in response to various cellular stimuli. At the outer mitochondrial membrane, Drp1 interacts with adaptor proteins, including Fis1, and triggers mitochondrial fission [28]. Drp1-Fis1 interaction is involved in oxidative stress-mediated mitochondrial fission, which further leads to apoptosis, cell death, and necrosis [29]. P110 is a specific inhibitor of Drp1-Fis1 interaction. Inhibition of mitochondrial fragmentation with P110 treatment restores mitochondrial membrane potential, reduces ROS levels, and restores oxidative respiratory capacity and ATP production [26].

The goal of this study is to elucidate the potential beneficial role of P110 in preserving neuronal health and delaying the progression of AD-related pathological processes using a human cell culture model. In the present study, we analyzed the expression of major genes and proteins that are involved in regulating Aβ production. We checked the expression APP, β-site APP cleaving enzyme 1 (BACE1), a disintegrin and metalloproteinase domain-containing protein 10 (ADAM10), and Klotho, the main genes involved in Aβ production and accumulation. Our study identified the neuroprotective potential of P110 on Aβ formation and documented improvement in mitochondrial function in SH-SY5Y human neuronal cells.

## 2. Materials and Methods

### 2.1. Cell Culture and P110 Treatment

SH-SY5Y human neuroblastoma cells (American Type Culture Collection, Manassas, VA, USA) were cultured in DMEM-F12 supplemented with 10% fetal calf serum (FCS), 2 mM L-glutamine and 50 μg per ml of penicillin-streptomycin at 37 °C in a 5% CO_2_ atmosphere. Cell culture media and supplementary reagents were obtained from Invitrogen (Grand Island, NY, USA). SH-SY5Y cells were plated at a density of 500,000 cells/mL. After 3 days, cells were treated with 1 µM and 10 µM of P110 (from Tocris, Bristol, UK). After 24 h or 48 h, cells were used and proceeded for further experiments.

The concentrations of P110 and duration of exposure were chosen based on previously published literature [28,30]. A literature search revealed that 1 µM for 24 h is widely used in cell culture and 1 µM is the physiologic concentration in a rat heart model [31]. We included a higher 10 µM concentration to check the differences in the results between 1 µM and 10 µM and to compensate for the short duration of the experiments compared to the prolonged drug exposure that would be possible in living organisms. 

### 2.2. Real-Time PCR

Total RNA was isolated after 24 h of P110 treatment using Trizol reagent and dissolved in nuclease-free water. The quantity of total RNA from each condition was measured by absorption at 260 and 280-nanometer wavelengths by ultraviolet spectrophotometry. Total cDNA was reverse transcribed from 1µg of total RNA. The reagents used in reverse transcription of total RNA into cDNA were 25 µM MgCl_2_, 10× PCR buffer (Invitrogen), 10 mM dNTP, 50 µM oligodT, 50 µM hexamers, RNAse inhibitor, and Moloney Murine Leukemia Virus reverse transcriptase enzyme. Samples were incubated in an Eppendorf thermocycler. The PCR running conditions were as follows: Step1: 25 °C for 5 min, Step 2: 42 °C for 60 min, Step 3: 80 °C for 5 min, Step 4: 4 °C upon completion. Total cDNA from each sample was then subsequently used for quantitative real-time PCR analysis on Light Cycler 480 (Roche Diagnostics, Rotkreuz, Switzerland) using FastStart SYBR Green Reagents Kit according to the manufacturer’s instructions. The standard real-time PCR protocol was followed using 40 PCR cycles. Statistical analysis of the quantitative real-time PCR was obtained by using the (2^−ΔΔCt^) method, normalized to housekeeping gene *GAPDH*. The list of primers used with their respective melting temperatures (Tm) is given in Table 1.

### 2.3. Western Blotting

Protein samples were collected from whole cell lysates using radioimmunoprecipitation assay (RIPA) lysis buffer (98% PBS, 1% Igepal, 0.5% sodium deoxycholate, 0.1% sodium dodecyl sulfate, supplemented with 10μL per ml of protease inhibitor cocktail (Sigma, Livonia, MI, USA) after 48 h of P110 treatment. Protein concentration was measured using the BCA Protein Assay Kit (Pierce Biotechnology Inc., Rockford, IL, USA). 8.3 µg of each protein sample was loaded, separated by 8–12% SDS-polyacrylamide gel electrophoresis (SDS-PAGE), and transferred to PVDF membranes. The blots were then subsequently blocked with 5% Milk in TBST for 1 h at room temperature and incubated in primary antibodies (APP: Abcam-ab32136, BACE1: Abcam-ab108394, Klotho: Abcam- ab181373, ADAM10: Abcam-ab124695, LRP1: Abcam-ab92544, Waltham, MA, USA) at 4 °C overnight. The membrane was then incubated with a secondary antibody (from GE Healthcare, anti-rabbit: NA934, anti-mouse: NA931) diluted in 5% Milk in TBST. The primary antibodies were used at 1:1000 dilution, and the secondary antibody was used at 1:5000 dilution. β-actin (Cell Signaling Technologies, CST3700, Danvers, MA, USA) was used as an internal control. Bound antibodies were visualized with an HRP-conjugated secondary antibody. The immunoreactive protein was detected using ECL western blotting detection reagents (Thermo Scientific™, Waltham, MA, USA, SuperSignal™ West Pico PLUS Chemiluminescent Substrate, Thermo Scientific™, Waltham, MA, USA) and the Bio-Rad ChemiDoc Touch Imaging System. Loading in each lane was validated using β-actin as an internal loading control. Quantification of protein blots was performed by using Image J software version 1.52a.

### 2.4. Mitotracker Staining

SH-SY5Y cells were stained with mitotracker dye that stains active mitochondria in live cells. Cells were seeded in a 6-well plate and treated with 1 µM and 10 µM of P110. After 48 h of P110 treatment, cells were washed with Hanks’ Balanced Salt Solution (HBSS) and stained with 250 µM of mitotracker (Thermo Fisher Scientific, M7512) for 20 min. After staining, cells were washed three times with HBSS. Images were captured at 20× using a fluorescence microscope, and total cell florescence was measured using ImageJ software.

### 2.5. Measurement of Intracellular Reactive Oxygen Species

To measure the intracellular reactive oxygen species levels, we used H2-DCFDA (Thermo Fisher Scientific, D399), a fluorescent dye. SH-SY5Y cells were seeded in 96-well and 6-well plates. Cells were treated with 1 µM and 10 µM of P110. After 48 h of P110 treatment, cells were washed with HBSS and incubated with 10 µM of H2-DCFDA for 30 min. After 30 min of incubation, cells were washed again with HBSS, and fluorescence intensity was measured at excitation wavelength 485 nm and emission wavelength 530 nm with a SpectraMax spectrophotometer. Images were captured at 20× using a fluorescence microscope, and total cell florescence was measured using ImageJ software version 1.52a.

### 2.6. Detection of Mitochondrial ROS Production

To detect mitochondrial ROS production, SH-SY5Y cells were stained with mitoSOX-Red (Thermo Fisher Scientific, M36007). Cells were seeded in 6 well plates and treated with P110 for 48 h. After 48 h of P110 incubation, cells were washed with HBSS and stained with 5 µM of mitoSOX-Red for 15 min. After 15 min of incubation, cells were washed again with HBSS, and images were captured at 20× using fluorescence microscopy. Total cell florescence was measured using ImageJ software version 1.52a.

### 2.7. Electron Microscopy

Cell culture samples were fixed in 2.5% glutaraldehyde buffered in 0.1 M sodium cacodylate buffer, pH 7.5, washed in 0.1 M sodium cacodylate buffer, and post-fixed and stained with 1% osmium tetroxide buffered in sodium cacodylate. Cells were then removed from the plastic culture dish, centrifuged, en-bloc stained with a saturated solution of uranyl acetate in 40% ethanol, dehydrated in a graded series of ethanol, infiltrated in propylene oxide with Epon epoxy resin (Embed812, Electron Microscopy Sciences, Hatfield, PA, USA), and embedded. The blocks were sectioned with a Reichert Ultracut microtome at 70 nm. Sections were picked up on 300 mesh copper grids, dried, and then post-stained with a 1% aqueous uranyl acetate followed by 0.5% aqueous lead citrate. Stained grids were examined on a Zeiss EM 900 transmission electron microscope retro-fitted with an SIA L3C digital camera (SIA, Duluth, GA, USA).

### 2.8. Statistics

Results are expressed as Mean ± SD. Statistical significance was analyzed by one-way ANOVA followed by Bonferroni’s multiple comparison test. Data were analyzed on Graph Pad Prism. Fold change was calculated between the control and treated groups. *p* values less than <0.05 were considered significant.

## 3. Results

### 3.1. P110 Treatment Regulates Aβ Formation in SH-SY5Y Cells

To investigate the role of P110 in amyloid β formation, we measured the expression of amyloid precursor protein (APP) and β-secretase-1 (BACE1) at both mRNA and protein levels in SH-SY5Y cells. P110 exposure in SH-SY5Y cells significantly increased APP mRNA levels at 1 µM, while mRNA levels of BACE1 were increased at both 1 µM and 10 µM concentrations (Figure 1A,B). However, protein levels of both APP and BACE1 were significantly reduced at 10 µM of P110 in SH-SY5Y cells (Figure 1C).

### 3.2. P110 Treatment Enhances Neuroprotective Effects in SH-SY5Y Cells

To measure the neuroprotective effects of P110 treatment in SH-SY5Y cells, we checked the expression of Low-density lipoprotein receptor-related protein 1 (LRP1) and synaptophysin at mRNA level and ADAM10, klotho, and LRP1 at protein levels. Real-time analysis showed a significant increase in mRNA level of synaptophysin and LRP1 at both 1 µM and 10 µM of P110 (Figure 2A,B). Further, the P110 treatment significantly increased protein levels of ADAM10 and klotho at 10uM in SH-SY5Y cells, but we did not find any difference in LRP1 at the protein level (Figure 2C).

### 3.3. P110 Treatment Increases Mitochondrial Function

Mitochondrial functioning was measured by measuring the expression of transcription factor A, mitochondrial (TFAM), and nuclear respiratory factor 1 (NRF1) levels, and staining of active mitochondria in live cells. Real-time PCR analysis showed that P110 treatment significantly increased mRNA levels of TFAM and NRF1 at both 1 µM and 10 µM concentrations in SH-SY5Y cells (Figure 3A,B). Further, P110-treated SH-SY5Y cells were stained with mitotracker dye to stain active mitochondria. Staining images showed a significant increase in active mitochondria at both 1 µM and 10 µM of P110, which was also confirmed by measuring the total cell fluorescence in live SH-SY5Y cells (Figure 3C,D).

### 3.4. P110 Treatment Reduces ROS Production

Total ROS production in SH-SY5Y cells after P110 treatment is determined by measuring intracellular and mitochondrial ROS production. For this, we stained P110-treated SH-SY5Y cells with H2DCFDA and mitoSOX dyes. Compared to control cells, H2DCFDA stained SH-SY5Y cells showed significantly reduced intracellular ROS production at both 1 µM and 10 µM of P110 treatment, as shown by measuring the total cell fluorescence and ROS levels in the cells (Figure 4A,B,D). We also measured mitochondrial ROS production using mitoSOX dye, which was also significantly reduced in SH-SY5Y cells after P110 treatment at both 1 µM and 10 µM concentrations (Figure 4A,C).

### 3.5. P110 Treatment Reduces Mitochondrial Fragmentation 

To determine the morphology of mitochondria, we did transmission electron microscopy (TEM) of SH-SY5Y cells after 48 h of P110 treatment. TEM images showed that SH-SY5Y cells treated with P110 at both 1 µM and 10 µM concentrations had less number of bigger size mitochondria as compared to control cells, which had more number of small mitochondria (Figure 5). Thus, it shows reduced mitochondrial fragmentation after P110 treatment.

## 4. Discussion

In this study, the neuroprotective role of P110 was explored using a human neuronal cell culture model. Exposure of SH-SY5Y cells to P110 reduced the expression of proteins involved in Aβ formation (APP, the precursor protein of Aβ peptide, and BACE1, the β-secretase cleavage enzyme required for the generation of amyloid-β peptides) and increased the levels of neuroprotective proteins (ADAM10, an α-secretase responsible of cleavage of APP through a non-amyloidogenic pathway and Klotho, involved in amyloid clearance) [32,33,34,35]. Further, P110 increased mitochondrial function and reduced ROS production. To our knowledge, this is the first study to evaluate the protective role of P110 in regulating Aβ formation and mitochondrial functioning in human neurons. 

The SH-SY5Y cell line is a widely used in vitro cellular model to screen and investigate novel therapeutics against neurodegenerative diseases, including AD [36]. It is very well reported that both undifferentiated and differentiated SH-SY5Y cells have been utilized for in vitro experiments to investigate neuronal health. Following the precedent of previously published studies which showed the importance of undifferentiated SH-SY5Y cells in exploring neuronal health [37,38,39], we also used the undifferentiated model of SH-SY5Y cells to evaluate the neuroprotective effect of P110. Emanuelsson et al. showed a neuroprotective effect of the oxysterols 27-hydroxycholesterol and 24-hydroxycholesterol against the toxicity of staurosporine in undifferentiated SH-SY5Y cells [37]. In another study conducted by Amar et al., the authors used an undifferentiated model of SH-SY5Y cells to assess the cellular and molecular responses of ethyl-parathion, an organophosphorous model compound, and its neurotoxic effects on SH-SY5Y cells [38]. 

Excessive accumulation of Aβ induces neuronal death, a critical hallmark in the pathogenesis of AD [40]. While the underlying process of cell death induction by Aβ is unknown, research is ongoing while also acknowledging the multifactorial nature of neuronal loss in AD. Carriba et al. showed a possible mechanism through which Aβ accumulation can promote neurodegeneration by reducing the expression of FAIM-L in neurons [41]. In this mechanism, abnormal accumulation of Aβ is the central event in neurotoxicity. Melatonin (N-acetyl-5-methoxytryptamine), a multifunctional neuroprotective agent that regulates inflammation, apoptosis, and oxidative stress, was shown in a rat model of sporadic Alzheimer’s Disease to slow reference memory loss. In these senescence-accelerated OXYS rats, oral melatonin administration reduced the Aβ_1–42_ and Aβ_1–40_ levels in the hippocampus and the frontal cortex [42]. 

BACE1, a neuronal enzyme responsible for the cleavage of APP to generate Aβ, is increased in both content and activity in the brains of AD patients [43,44]. The importance of BACE1 is supported by different in vitro and in vivo studies showing that its deletion abolishes Aβ formation, and thus, it has served as a therapeutic target in AD [45,46,47]. Li et al. demonstrated the role of miR-15b in the reduction of BACE1 and APP expression in a human cell model of AD [48]. Our study very nicely corroborates their earlier reports. The present study showed increased mRNA levels of BACE1 and APP after P110 treatment. Despite increasing these mRNA levels, P110 has an anti-amyloidogenic effect because it reduces the formation of Aβ by lowering the protein levels of APP and BACE1. Due to post-transcriptional modification, protein expression is not necessarily proportional to mRNA level. Thus, from the above-mentioned results, we found that P110 showed its effect on translational levels only and reduced the protein levels of APP and BACE1 in SH-SY5Y cells. Since the physiologic effects are exerted by the protein, we assert that these are the relevant data. The lack of concordance of mRNA and protein is a very common issue in the complex environment of living cells and organisms [49,50,51].

Parallel to the anti-amyloidogenic effect, P110 has the potential to provide neuroprotection. To our knowledge, this is the first study to show improvements in the expression of genes associated with neuroprotection after P110 treatment. Klotho has a beneficial role in preventing neuronal damage and offering neuroprotection in several neurodegenerative diseases [52,53]). The increased level of Klotho protein after P110 exposure in neurons is in accord with the previous literature showing the upregulation of Klotho in the neuroprotective effect of ligustilide against AD [54]. ADAM10 is another important molecule that serves as a potential therapeutic target to slow the progression of AD. ADAM10 has been identified as a main α-secretase responsible for cleavage of APP through a non-amyloidogenic pathway, thus reducing the production and accumulation of Aβ [55]. Our results showing reduced Aβ formation as a result of lower levels of APP and BACE1 and increased levels of ADAM10 with P110 treatment of SH-SY5Y represent a promising strategy in the treatment of AD. However, the full mechanism of ADAM10 activation and reduction of APP levels is not fully elucidated and requires additional experiments to create a complete picture. 

Several studies have confirmed the detection of mitochondrial dysfunction in the course of AD [14,16]. Reports showed the potential consequences of alteration in the expression of electron transport chain proteins in AD progression [56,57,58]. Evidence from cell culture and animal models showed a direct effect of tau and Aβ on mitochondrial dysfunction [59,60]. Rhein et al. found that impairment of the OXPHOS system in triple transgenic AD mice was dependent on tau and Aβ [60]. Accumulation of Aβ within the mitochondrial matrix provides a direct link between mitochondrial toxicity and AD progression [61]. A study conducted by Manczak et al. in a transgenic mouse model of AD demonstrated that accumulation of Aβ in mitochondria triggered hydrogen peroxide production and reduced cytochrome c activity [62].

P110 is a selective inhibitor of mitochondrial fission and fragmentation. It inhibits excess activation of mitochondrial protein Drp1 by inhibiting the interaction of Drp1 and Fis1 in mitochondria under stress conditions in different in vitro and in vivo models of various neurodegenerative diseases, including Parkinson’s, Huntington’s, and Alzheimer’s diseases [26,28,63]. P110 inhibits microglial cell activation by inhibiting mitochondrial fission and pro-inflammatory cytokine release, thus providing neuronal protection in neurodegenerative diseases [64]. Transmission electron microscopy data confirmed the above role of P110 in our study. In SH-SY5Y neurons, mitochondrial area was increased with reduced mitochondrial fragmentation after P110 treatment. Using murine and cell culture models, Joshi et al. showed that inhibition of Drp1 with P110 was associated with improved mitochondrial function and attenuation of structural and functional mitochondrial defects caused by Aβ [28]. 

Oxidative stress is another major factor that plays a crucial role in the progression of AD. Oxidative stress markers are increased in the brains of patients with AD. Therefore, an anti-oxidative agent is required to maintain an equilibrium between free radical production and antioxidant defense [65]. A small bifunctional molecule, spin-labeled fluorene (SLF) reduced Aβ accumulation in MC65 cells and N2A neuronal cells. Confocal microscopy images showed decreased Aβ induced ROS signals after SLF exposure [66]. Loss of ATP synthesis and accumulation of ROS are involved in excessive mitochondrial fragmentation or fission and result in dysfunctional mitochondria and DRP1 is a driver of ROS production [26,67]. Interestingly, Bordt et al. demonstrate that Mdivi-1 inhibits Complex I-dependent Mitochondrial O2 Consumption and ROS production in neurons [68]. Consistent with the above reports, we also observed increased mitochondrial function and reduced ROS production in neurons incubated with P110. 

Our study reveals an inhibitory effect of P110 on modulation of Aβ formation, which further leads to improvement of neuronal health in human SH-SY5Y neuroblastoma cells. P110 treatment reduced oxidative stress by reducing ROS production in SH-SY5Y cells. Thus, our study reveals a possible mechanistic link between reduced Aβ accumulation and improved mitochondrial health. This suggests that improvement in mitochondrial health after P110 treatment would be due, at least in part, to reduced Aβ accumulation in mitochondria. Our further investigation will focus on the mechanism of how the accumulation of Aβ in mitochondria and other metabolic disruptions that compromise mitochondrial bioenergetics are involved in the progression of AD.

The limitations of this study are related to its cell culture approach. The physiological relevance of this work is not fully validated and requires further studies to analyze whether P110 regulation of Aβ accumulation and neuronal health also occurs in vivo. Nonetheless, the use of human cell lines avoids some of the major issues that occur in murine studies where manipulation of transgenic mice yields benefits that do not translate to humans. It is important to use both types of approaches in a complementary fashion to move stepwise toward human clinical trials [69]. Future experiments will reinforce the role of P110 in counteracting the deposition of Aβ and reducing anomalies in mitochondria in primary human neurons and other, more complex cell systems.

In addition, we recognize that this data is preliminary. The study addresses multiple aspects, including the amyloidogenic pathway of AD, synaptic proteins, and mitochondrial function, which make it challenging to draw definitive conclusions, particularly with regard to causal relationships. The changes in the amyloidogenic pathway genes cannot be linked to the improved mitochondrial function. Additional studies are warranted to uncover the mechanisms underlying the documented changes in mitochondrial activity and ROS generation. However, consistent with our results, the drug metformin has been found to decrease neuronal loss in a mouse model of diabetes with impaired memory by inhibiting Drp1, causing reduced mitochondrial fission and a decline in ROS production [70]. Other studies in mice and in cell culture also link Aβ to increased Drp1 activity and mitochondrial fragmentation [71]. A murine study showed that the polyphenolic compound baicalin could counteract Aβ-induced neuronal dysfunction in mice by stimulating phosphorylation of Drp1 at serine 637, which renders it inactive with mitochondria. The baicalin treatment led to a reversal of mitochondrial dysfunction and fragmentation via Drp1 phosphorylation [72,73].

## 5. Conclusions

In this study, we have shown that P110 has anti-amyloidogenic properties in human SH-SY5Y neuronal cells. P110 benefits manifested at a translational level as favorable effects were at the level of protein rather than mRNA. Our results indicate that P110 might be useful in attenuating Aβ formation, enhancing neuroprotection, and supporting mitochondrial health. Additional studies will clarify this benefit, which may have clinical application in conjunction with other AD treatments to improve prognosis in this complex, multifactorial neurodegenerative disorder.

## Figures and Tables

**Figure 1 life-13-02156-f001:**
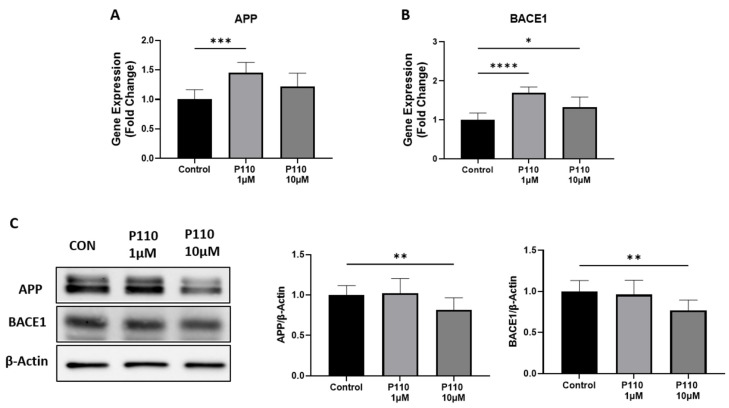
P110 exposure regulates APP and BACE1 expression in SH-SY5Y cells. (**A**,**B**) Real-time PCR analysis of APP and BACE1 in SH-SY5Y cells after P110 treatment. GAPDH was used as an internal control. Data was represented in fold difference, *n* = 6–9. (**C**) Western blot analysis of APP and BACE1 in SH-SY5Y cells after P110 treatment. Densitometry of representative blot was normalized with β-actin. Data was represented in fold difference, *n* = 6–9. **** *p* < 0.0001, *** *p* < 0.001, ** *p* < 0.01, and * *p* < 0.05, based on a one-way ANOVA followed by Bonferroni multi-comparison tests.

**Figure 2 life-13-02156-f002:**
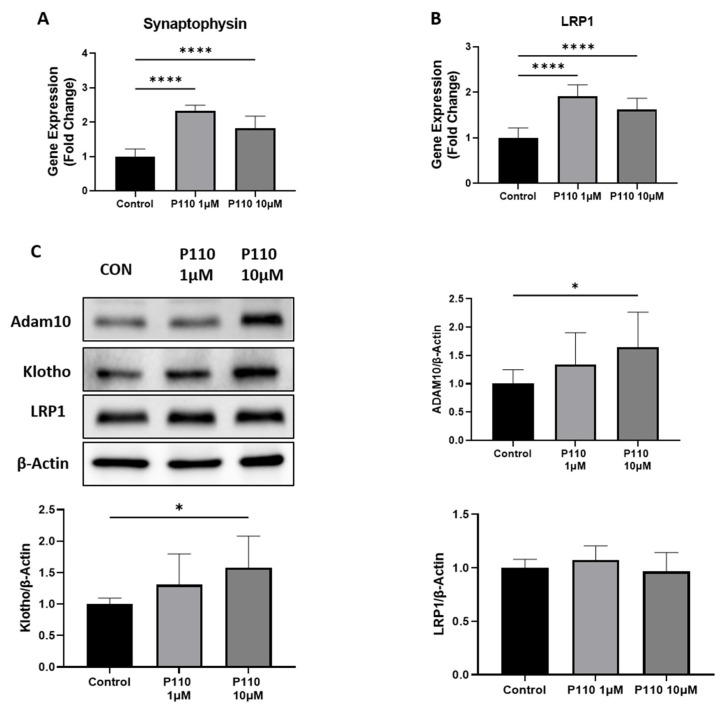
P110 exposure increases expression of genes involved in neuroprotection. (**A**,**B**) Real-time PCR analysis of Synaptophysin and LRP1 in SH-SY5Y cells after P110 treatment. GAPDH was used as an internal control. Data was represented in fold difference, *n* = 6–9. (**C**) Western blot analysis of Synaptophysin and LRP1 in SH-SY5Y cells after P110 treatment. Densitometry of representative blot was normalized with β-actin. Data was represented in fold difference, *n* = 6–9. **** *p* < 0.0001 and * *p* < 0.05, based on a one-way ANOVA followed by Bonferroni multi-comparison tests.

**Figure 3 life-13-02156-f003:**
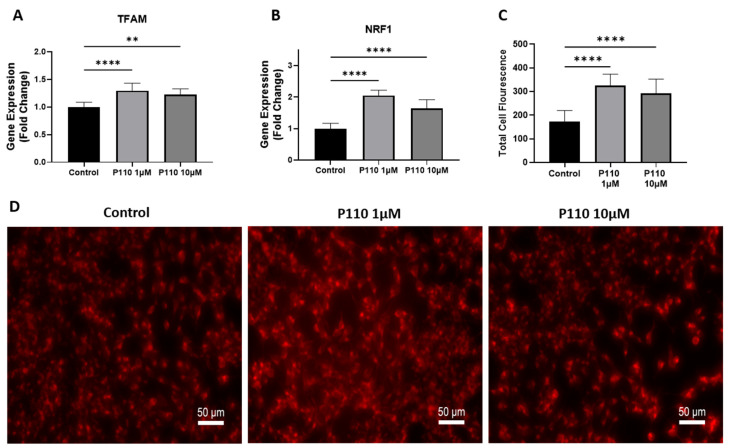
P110 exposure improves mitochondrial function in SH-SY5Y cells. (**A**,**B**) Real-time PCR analysis of TFAM and NRF1 in SH-SY5Y cells after P110 treatment. GAPDH was used as an internal control. (**C**,**D**) Total cell fluorescence and microscopic images of live SH-SY5Y cells stained with 250 µM of mitotracker dye after P110 treatment. Scale bar: 50 µm. Data was represented in fold difference. *n* = 6–9, **** *p* < 0.0001 and ** *p* < 0.01, based on a one-way ANOVA followed by Bonferroni multi-comparison tests.

**Figure 4 life-13-02156-f004:**
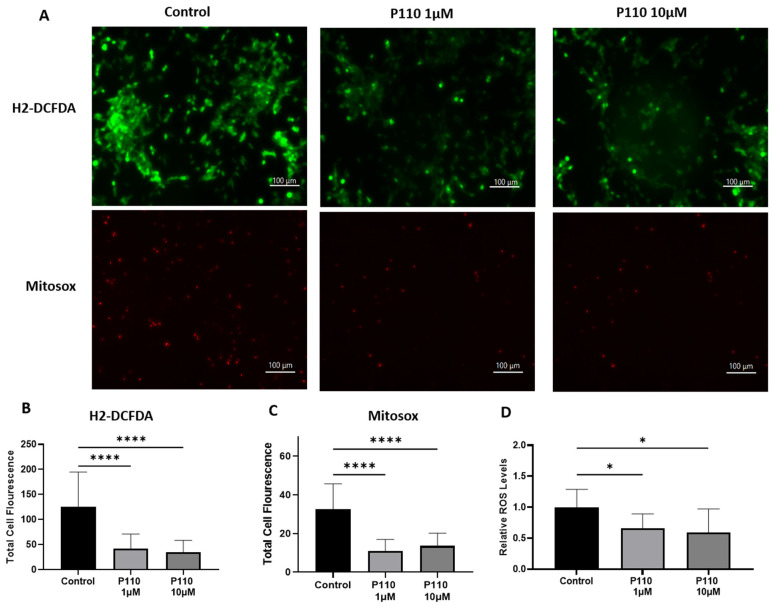
P110 exposure reduces overall ROS production in SH-SY5Y cells. (**A**) Microscopic images of P110 treated SH-SY5Y cells after H2-DCFDA and mitosox staining. Scale bar: 100 µm. (**B**,**C**) Total cell fluorescence measured from microscopic images using ImageJ software. (**D**) Measurement of ROS levels in H2-DCFDA stained SH-SY5Y cells after P110 treatment using SpectraMax spectrophotometer Data was represented in fold difference. *n* = 6–9, **** *p* < 0.0001 and * *p* < 0.05, based on a one-way ANOVA followed by Bonferroni multi-comparison tests.

**Figure 5 life-13-02156-f005:**
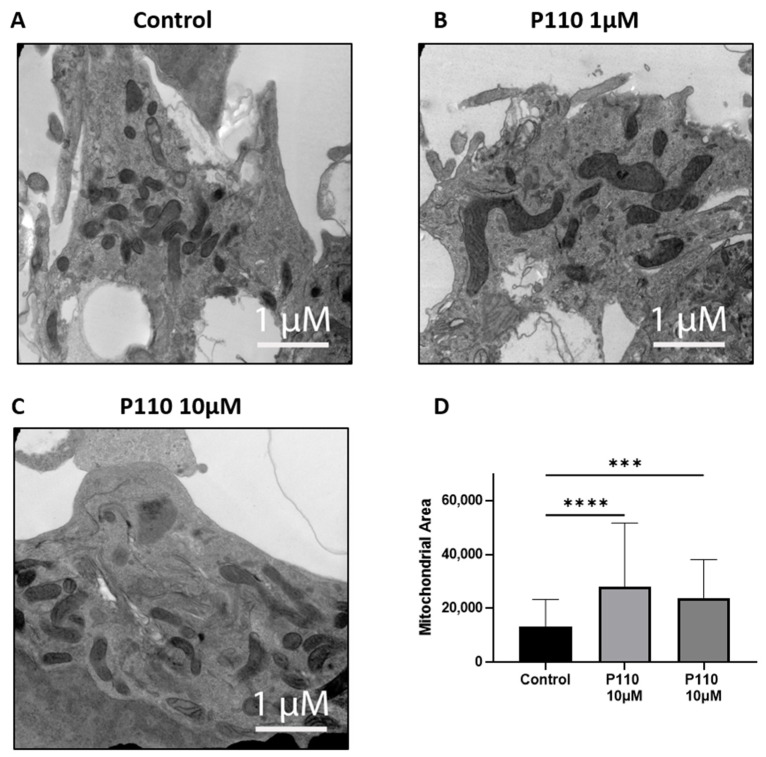
P110 exposure reduces mitochondrial fission and increases mitochondrial area in SH-SY5Y cells. (**A**–**C**) Representative transmission electron microscopy images of P110 treated SH-SY5Y cells. Scale bar: 1 μM (**D**) Measurement of mitochondrial area using ImageJ software. **** *p* < 0.0001 and *** *p* < 0.001, based on a one-way ANOVA followed by Bonferroni multi-comparison tests.

**Table 1 life-13-02156-t001:** Human primer sequences with Tm for real-time PCR.

Primer (Tm)	Forward Sequence	Reverse Sequence
GAPDH (62 °C)	ACCATCATCCCTGCCTCTAC	CCTGTTGCTGTAGCCAAAT
APP (62 °C)	TTTGGCACTGCTCCTGCT	CCACAGAACATGGCAATC
BACE-1 (62 °C)	GCAGGGCTACTACGTGGAGA	CAGCACCCACTGCAAAGTTA
TFAM (63 °C)	AAGATTCCAAGAAGCTAAGGGTGA	CAGAGTCAGACAGATTTTTTCCAGTTT
SYNAPTOPHYSIN (65 °C)	CTGCAATGGGTCTTCGCCA	ACTCTCGGTCTTGTTGGC
LRP-1 (63 °C)	ATGGGCAGATCCCAAAGGTG	CAGTCATTGTCATTGTCGCATCT
NRF-1 (63 °C)	GGCACTGTCTCACTTATCCAGGTT	CAGCCACGGCAGAATAATTCA

## Data Availability

The full dataset is available from the corresponding author upon motivated request.

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
