# Peer review of "Therapeutic Potential of P110 Peptide: New Insights into Treatment of Alzheimer’s Disease"

_life, 2023, doi:10.3390/life13112156_

Round 1

Reviewer 1 Report

Comments and Suggestions for Authors

The manuscript “Preclinical Evaluation of a Small Molecule Inhibitor of Drp1 as a Treatment for Alzheimer’s Disease” would investigate the neuroprotective role of P110, a small peptide able to act on mitochondrial dynamics, in SH-SY5Y cells, a cell line commonly used to test pharmacological effects of potential drugs. The authors showed that P110 is able to attenuate the deposition of beta-amyloid in SH-SY5Y cells by an increase of expression of APP and BACE1 mRNA and by a decrease of protein expression of APP and BACE1; contextually, an increase of activity of mitochondria and a reduction of ROS was observed.

The introduction well explains the backgrounds of the experiments which are reported in the manuscripts and the methods and results are clear and well written. The discussion well justifies the results on the basis of current literature.

However, the authors considered many times Sh-SY5Y as neuronal model, not considering the tumoral nature of the cells, which also include an epithelioid component that may perturb the neuronal origin of this cell line. In literature, many methods to differentiate SH-SY5Y cells in neuronal-like cells are reported, such as the treatment with a combination of retinoic acid (RA) and BDNF. Did you consider doing the same experiments on differentiate cells to verify if the same effects will be observed in a cell model that is more similar to the effective target of P110? Moreover, the title of the manuscript (i.e., “pre-clinical”) let us to think that you use a patient-derived cell model that can resemble the in vivo effects. I think that more experiments will enforce the role of P110 in contrasting the deposition of beta amyloid and to reduced anomalies in mitochondria in a more adherent cell system.

Author Response

We thank the reviewer for thoroughly scrutinizing our manuscript. As requested, we have revised the manuscript and addressed the specific comments of the reviewer. The revised sections are delineated in red in a marked copy of the manuscript text.

Below, we provide a point-by-point response to the reviewer’s comments.

Reviewer # 1 Comments and Responses

  • COMMENT #1: However, the authors considered many times Sh-SY5Y as neuronal model, not considering the tumoral nature of the cells, which also include an epithelioid component that may perturb the neuronal origin of this cell line. In literature, many methods to differentiate SH-SY5Y cells in neuronal-like cells are reported, such as the treatment with a combination of retinoic acid (RA) and BDNF. Did you consider doing the same experiments on differentiate cells to verify if the same effects will be observed in a cell model that is more similar to the effective target of P110? Moreover, the title of the manuscript (i.e., “pre-clinical”) let us to think that you use a patient-derived cell model that can resemble the in vivo effects. I think that more experiments will enforce the role of P110 in contrasting the deposition of beta amyloid and to reduced anomalies in mitochondria in a more adherent cell system.

RESPONSE: We have expanded upon our explanation for use of SH-SY5Y cells (page 9, lines 264-274). The term “pre-clinical” is commonly used to denote a study that does not use human subjects, but could lead to human trials. We agree that more experiments are warranted and will plan to perform them in long-term future studies.

We now include this paragraph: It is very well reported that both undifferentiated and differentiated SH-SY5Y cells have been utilized for in vitro experiments to investigate neuronal health. Following the precedent of previously published studies which showed the importance of undifferentiated SH-SY5Y cells in exploring neuronal health (References 37, 38, and 39), we also used the undifferentiated model of SH-SY5Y cells to evaluate the neuroprotective effect of P110. Emanuelsson et. al. showed a neuroprotective effect of the oxysterols 27-hydroxycholesterol and 24- hydroxycholesterol against the toxicity of staurosporine in undifferentiated SH-SY5Y cells (Reference 37). In another study conducted by Amar et. al., the authors used an undifferentiated model of SH-SY5Y cells to assess the cellular and molecular responses of ethyl-parathion, an organophosphorous model compound and its neurotoxic effects on SH-SY5Y cells (Reference 38).

We have made the need for more experiments clear by elaborating in our description of limitations (lines 365-367).

We now include this statement: Future experiments will reinforce the role of P110 in counteracting the deposition of Aβ and reducing anomalies in mitochondria in primary human neurons and other, more complex cell systems.

We thank the reviewer and believe that the manuscript is improved as a result of their input.  We hope you will agree, and decide in favor of accepting our report at this time.

Reviewer 2 Report

Comments and Suggestions for Authors

Knowing that mitochondrial dysfunction is associated with Alzheimer disease (AD) and that the interaction between Drp1 and Fis1 induces mitochondrial fission and cellular oxidative stress, the authors have tested P110 – a peptide that targets specifically Drp1-Fis1 interaction- on a neuronal cell culture model for its effects on the expression of genes and proteins associated with the amyloidogenic pathway of AD as well as synaptic proteins and markers of mitochondrial function and integrity. I found that this study is too preliminary, and the results inadequately interpreted.

Specific comments.

1.     The background is missing information to understand the rationale of the study. Why did you choose to analyse APP, BACE1, ADAM10 and KLOTHO? You should indicate that you analysed major genes and proteins involved in the molecular processes that control Ab amyloid production.  APP is the precursor to Ab, BACE1 is the rate-determining cleaving enzyme in Ab production, ADAM10 (alpha-secretase) is the alternative cleaving enzyme that shifts APP towards a non-amyloid genic pathway, and KLOTHO has been shown to improve amyloid clearance.

2.     The source of the antibodies is missing from the Materials and Methods section.

3.     The choice of P110 concentrations used in the study and the time of exposure to the cells (24 h or 48 h) need to be justified. Have you done dose-response and cell viability preliminary studies to determine optimal conditions?

4.     How do you interpret the discrepancy between the results of mRNA and western blot for APP and BACE1?

5.     Line 231: how do you know that P110 decreased Abeta in this study? You cannot draw such a conclusion just based on a marginal decrease in its precursor protein. You would need to measure Abeta peptides in the culture media.

Author Response

We thank the reviewer for thoroughly scrutinizing our manuscript. As requested, we have revised the manuscript and addressed the specific comments of the reviewer. The revised sections are delineated in red in a marked copy of the manuscript text.

Below, we provide a point-by-point response to the reviewer’s comments.

Reviewer # 2 Comments and Responses

1) COMMENT #1:    The background is missing information to understand the rationale of the study. Why did you choose to analyse APP, BACE1, ADAM10 and KLOTHO? You should indicate that you analysed major genes and proteins involved in the molecular processes that control Ab amyloid production.  APP is the precursor to Ab, BACE1 is the rate-determining cleaving enzyme in Ab production, ADAM10 (alpha-secretase) is the alternative cleaving enzyme that shifts APP towards a non-amyloid genic pathway, and KLOTHO has been shown to improve amyloid clearance.

RESPONSE: We have incorporated the above-mentioned information in the introduction and discussion section of the main text (page 2, lines 78-82, page 9, lines 255-260).

2) COMMENT #2:  The source of the antibodies is missing from the Materials and Methods section.

RESPONSE: We have incorporated the antibodies name and catalogue numbers in the Material and Methods section of main manuscript (page 3, lines 126-129).

3) COMMENT #3:  The choice of P110 concentrations used in the study and the time of exposure to the cells (24 h or 48 h) need to be justified. Have you done dose-response and cell viability preliminary studies to determine optimal conditions?

RESPONSE: We have now added a paragraph describing how we made the decision on concentration (page 2, lines 94-99).

We now include this paragraph: The concentrations of P110 and duration of exposure were chosen based on previously published literature (References 28 and 30). A literature search revealed that 1µM for 24 hrs is widely used in cell culture and 1µM is the physiologic concentration in a rat heart model (Reference 31). We included a higher 10µM concentration to check the differences in the results between 1µM and 10µM and to compensate for the short duration of the experiments compared to the prolonged drug exposure that would be possible in living organisms.

As we did not find any difference in protein levels at 24 hrs, we extended our study to 48 hrs for measurements of protein levels. We did not perform dose-response and cell viability studies as the dose is very well reported in previously published literature (References 28 and 30).

4) COMMENT #4:    How do you interpret the discrepancy between the results of mRNA and western blot for APP and BACE1?

RESPONSE: We have included in the discussion section of the main manuscript a paragraph explaining our thoughts on this discrepancy (page 10, lines 294-300).

We now include this paragraph: Due to post-transcriptional modification, protein expression is not necessarily proportional to mRNA level. Thus, from the above-mentioned results, we found that P110 showed its effect on translational levels only and reduced the protein levels of APP and BACE1 in SH-SY5Y cells. Since the physiologic effects are exerted by the protein, we assert that these are the relevant data. The lack of concordance of mRNA and protein is a very common issue in the complex environment of living cells and organisms (References 49-51).

5) COMMENT #5: Line 231: how do you know that P110 decreased Abeta in this study? You cannot draw such a conclusion just based on a marginal decrease in its precursor protein. You would need to measure Abeta peptides in the culture media.

RESPONSE: We have modified this and changed the wording for precision to ‘reduced the expression of proteins  involved in Aβ formation”

We thank the reviewer and believe that the manuscript is improved as a result of their input.  We hope you will agree, and decide in favor of accepting our report at this time.

Reviewer 3 Report

Comments and Suggestions for Authors

Review of a manuscript “Preclinical Evaluation of a Small Molecule Inhibitor of Drp1 as a Treatment for Alzheimer’s Disease “ by Ankita Srivastava submitted to “Life”

Alzheimer’s disease is a prevalent neurodegenerative disease and the cause of dementia associated with progressive memory loss and cognitive decline. In spite of much research in the area there is no medication that reverses the disease process. The authors carried out cell culture investigation to assess the neuroprotective effect of a seven-amino acid peptide P110 on amyloid-β accumulation and functions of mitochondria. This is a very important area of biomedical research, and the results presented in the manuscript will be interesting for the readership of the journal.

The following corrections and additions should be made.

Introduction:

Line37. “Aβ plaques consist of deposition of misfolded Aβ peptide, a cleavage product of amyloid precursor protein (APP).” After this sentence the authors should add a reference on a recent review: “Controversial properties of amyloidogenic proteins and peptides: new data in the COVID era. Biomedicines 2023, 10, 11(4):1215. DOI: 10.3390/biomedicines11041215

Line 58. “An adequate in vitro model is key for preclinical studies of the pathophysiological mechanism of disease and the impact of potential therapies.” The sentence is unclear and should be corrected as follows:” An adequate in vitro model is a key for preclinical studies of the pathophysiological mechanism of disease and for the assessment of  the impact of potential therapies.”

Materials and Methods:

Line 86: ” SH-SY5Y cells were plated at a density of 500,000 cells/mL” This is repeated twice in Materials and Method.

Line 89. “2.2. Real-Time PCR” Conditions of qRT-PCR should be given (temperature, number of cycles, etc.)

Line 93: “DNA was synthesized from 1µg of total RNA.” Details of reverse transcription should be given.

Line 109. “The membrane was then incubated with secndary antibodies diluted in 5% Milk in TBST.” The sentence should be corrected as follows :” The membrane was then incubated with secondary antibodies diluted in 5% Milk in TBST.” Dilution of antibodies should be given.

Results

Figure 3.

Line 196. “P110 exposure improves mitochondrial dysfunctioning in SH-SY5Y cells”. This is awkward sentence, should be rewritten to better understand what does it mean “improves dysfunctioning”; dysfunctioning means dysfunction?

In D scale bar should be added.

Figure 4, A. scale bar should be added.

Figure 5 A-C. scale bar should be added.

Discussion

It would be beneficial if the authors extrapolate their data and present their opinion, at least hypothetical, how P110 would regulate Aβ accumulation and neuronal health in mammalian and human tissues in vivo.

Author Response

We thank the reviewer for thoroughly scrutinizing our manuscript. As requested, we have revised the manuscript and addressed the specific comments of the reviewer. The revised sections are delineated in red in a marked copy of the manuscript text.

Below, we provide a point-by-point response to the reviewer’s comments.

Reviewer # 3 Comments and Responses

  • COMMENT #1: Introduction: “Aβ plaques consist of deposition of misfolded Aβ peptide, a cleavage product of amyloid precursor protein (APP).” After this sentence the authors should add a reference on a recent review: “Controversial properties of amyloidogenic proteins and peptides: new data in the COVID era. Biomedicines 2023, 10, 11(4):1215. DOI: 10.3390/biomedicines11041215

RESPONSE: We have incorporated the above-mentioned reference in the main text.

  • COMMENT #2: Introduction: Line 58. “An adequate in vitro model is key for preclinical studies of the pathophysiological mechanism of disease and the impact of potential therapies.” The sentence is unclear and should be corrected as follows:” An adequate in vitro model is a key for preclinical studies of the pathophysiological mechanism of disease and for the assessment of the impact of potential therapies.”

RESPONSE: We have corrected this.

  • COMMENT #3: Materials and Methods:
  1. Line 86: ” SH-SY5Y cells were plated at a density of 500,000 cells/mL” This is repeated twice in Materials and Method.

Response: We have corrected this.

  1. Line 89. “2.2. Real-Time PCR” Conditions of qRT-PCR should be given (temperature, number of cycles, etc.)

Response: Conditions have been added as requested (page 3, lines 112-115).

  1. Line 93: “DNA was synthesized from 1µg of total RNA.” Details of reverse transcription should be given.

Response: Details have been added as requested (page 3, lines 104-109).

  1. Line 109. “The membrane was then incubated with secndary antibodies diluted in 5% Milk in TBST.” The sentence should be corrected as follows :” The membrane was then incubated with secondary antibodies diluted in 5% Milk in TBST.” Dilution of antibodies should be given.

Response: Dilution has now been included (page 3, lines 129-130).

  • COMMENT #4: Results:
  1. Figure 3, Line 196. “P110 exposure improves mitochondrial dysfunctioning in SH-SY5Y cells”. This is awkward sentence, should be rewritten to better understand what does it mean “improves dysfunctioning”; dysfunctioning means dysfunction?

Response: Sentence has been reworded for clarity.

  1. In D scale bar should be added.

Response: Scale bar has been added.

  1. Figure 4, A. scale bar should be added.

Response: Scale bar has been added.

  1. Figure 5 A-C. scale bar should be added.

Response: Scale bar has been added.

  • COMMENT #5: Discussion: It would be beneficial if the authors extrapolate their data and present their opinion, at least hypothetical, how P110 would regulate Aβ accumulation and neuronal health in mammalian and human tissues in vivo.

RESPONSE: We have incorporated a paragraph in the discussion section (page 11, lines 350-358).

Here is the added material:  Our study reveals an inhibitory effect of P110 on modulation of Aβ formation, which further leads to improvement of neuronal health in human SH-SY5Y neuroblastoma cells. P110 treatment reduced oxidative stress by reducing ROS production in SH-SY5Y cells. Thus, our study reveals a possible mechanistic link between reduced Aβ accumulation and improved mitochondrial health. This suggests that improvement in mitochondrial health after P110 treatment would be due, at least in part, to reduced Aβ accumulation in mitochondria. Our further investigation will focus on the mechanism of how accumulation of Aβ in mitochondria and other metabolic disruptions that compromise mitochondrial bioenergetics are involved in the progression of AD.

We thank the reviewer and believe that the manuscript is improved as a result of their input.  We hope you will agree, and decide in favor of accepting our report at this time.

Round 2

Reviewer 1 Report

Comments and Suggestions for Authors

The Authors answered adequately to my requests of revision

Author Response

Comment: The Authors answered adequately to my requests of revision.

Response: We sincerely appreciate your valuable comments and suggestions. Your constructive feedback helped us in improving the quality of the manuscript.
